# Bifurcation, chaos, and stability analysis to the second fractional WBBM model

**Mohammad Safi Ullah**[1,2]*, **M. Zulfikar Ali**[2], **Harun-Or Roshid**[3]

**1** Department of Mathematics, Comilla University, Cumilla, Bangladesh, **2** Department of Mathematics, University of Rajshahi, Rajshahi, Bangladesh, **3** Department of Mathematics, Pabna University of Science and Technology, Pabna, Bangladesh

* safi.ru1985@gmail.com

**Data Availability Statement:** All relevant data are within the manuscript.

**Funding:** The author(s) received no specific funding for this work.

## Abstract

This manuscript investigates bifurcation, chaos, and stability analysis for a significant model in the research of shallow water waves, known as the second 3D fractional Wazwaz-Benjamin-Bona-Mahony (WBBM) model. The dynamical system for the above-mentioned nonlinear structure is obtained by employing the Galilean transformation to fulfill the research objectives. Subsequent analysis includes planar dynamic systems techniques to investigate bifurcations, chaos, and sensitivities within the model. Our findings reveal diverse features, including quasi-periodic, periodic, and chaotic motion within the governing nonlinear problem. Additionally, diverse soliton structures, like bright solitons, dark solitons, kink waves, and anti-kink waves, are thoroughly explored through visual illustrations. Interestingly, our results highlight the importance of chaos analysis in understanding complex system dynamics, prediction, and stability. Our techniques' efficiency, conciseness, and effectiveness advance our understanding of this model and suggest broader applications for exploring nonlinear systems. In addition to improving our understanding of shallow water nonlinear dynamics, including waveform features, bifurcation analysis, sensitivity, and stability, this study reveals insights into dynamic properties and wave patterns.

## 1 Introduction

Solitons are solitary wave solutions that preserve their structure and amplitude while traveling through a system without dispersing or losing energy due to dispersion or dissipation [1]. They are fascinating entities that emerge from the delicate interplay between dispersion and nonlinearity [2], and they have been observed and studied extensively in different disciplines, including fluid dynamics [3], plasma physics [4], condensed matter physics [5], and optics [6]. Consequently, the attraction of soliton solutions in the field of mathematical physics is considerable [7, 8]. Many distinguished nonlinear models provide evidence of the presence of these soliton outcomes like the Jimbo-Miwa structure [9], Phi-4 equation [10], the extended Boussinesq nonlinear framework [11], the Nizhnik-Novikov-Veselov model [12], the ISLW system [13], the 3D fractional WBBM structure [14], Fokas-Lenells system [15], and various others [16–18]. Nonlinear models can be solved using a variety of effective methods, and soliton

**Competing interests:** The authors have declared that no competing interests exist.

outcomes can be obtained. This list includes some techniques, such as the Kudryashov technique [19], the unified process [20], the $\exp_a$ function approach [21], the modified Sardar sub-equation algorithm [22], the Hirota bilinear process [23], and various others [24–26].

Bifurcation analysis is a robust technique widely used to explore dynamic structures, which has significant implications in various fields [27, 28]. The bifurcation process, announced by Liu and Li in 2002 [29], is a necessary tool for probing the dynamics of nonlinear models. Particularly efficient at scrutinizing the bifurcation nature and deriving precise soliton outcomes, this method investigates how altering system parameters influences its qualitative behavior [30]. By studying bifurcation, investigators gain insights into changes between stable and unstable situations or chaotic dynamics. This manuscript aims to explore novel waveforms and bifurcation analysis of the 2nd fractional WBBM nonlinear structure [31].

The synopsis of the existent investigation is as follows: In Section 2, an explanation of the non-integer order derivative and its basic properties is presented. Section 3 delves into the description and ordinary differential form of the second fractional 3D-WBBM system. Bifurcation analysis is thoroughly described in Section 4, while Section 5 addresses the chaotic activities of the adopted nonlinear structure. The sensitivity examination of the projected nonlinear problem is contained in Section 6, and Section 7 shows the appearance of bright and dark solitons in the model. Section 8 contains the graphical representations conducted for the study. In Section 9, the novelty of the outcomes is addressed, demonstrating original findings attained through the procedures recommended for Eq (7), which have not been previously stated. Finally, Section 10 captures the conclusions drawn from this document.

## 2 Conformable derivative and its renowned characteristics

To understand the dynamic processes in function development, the fractional derivative is compared with the numerical derivative, which illustrates their overarching connections. Fractional calculus offers a versatile approach applicable to various domains, such as hydrodynamics, applied mathematics, fluid mechanics, quasi-chaotic dynamical systems, system validation, finance, unpredictable fluid infrastructures, and research methodologies. Additionally, it extends to diverse topics like optical fibers, solid-state biological processes, environmental studies, and theoretical electrical control, among others. A fractional derivative provides a clear explanation for the nonlocal properties of mathematical representations that differ from conventional calculus, which focuses solely on the present state of matter. Over time, various fractional derivatives have been devised to depict significant physical phenomena [32]. Riemann-Liouville derivatives modified by Jumarie [33], conformable derivatives of Atangana [34], their beta derivative [35], and derivatives of Caputo [36] provide a better representation than integer-order derivatives. These derivatives find application across diverse domains in contemporary science and engineering.

Consider $w : [0, \infty] \to \mathbb{R}$, then the conformable fractional derivative of $w(t)$ with order $\gamma \in (0, 1]$ is denoted by $D_t^\gamma w(t)$ and is defined by [37]:

$$D_t^\gamma w(t) = \lim_{\epsilon \to 0} \frac{w(t + \epsilon t^{1-\gamma}) - w(t)}{\epsilon}, t > 0. \tag{1}$$

Here we include some renowned characteristics of fractional derivatives of specified order $\gamma \in (0, 1]$ w. r. to $t > 0$. If $w(t) = w$ and $r(t) = r$ are any real functions, then
(i) $D_t^\gamma(a) = 0, \forall$ constant function $w(t) = a$.
(ii) $D_t^\gamma t^n = nt^{n-\gamma}, \forall n \in \mathbb{R}$
(iii) $D_t^\gamma(bw) = bD_t^\gamma w, \forall n \in \mathbb{R}$.

(iv) $D_t^\gamma(cw + dr) = cD_t^\gamma w + dD_t^\gamma r, \forall c, d \in \mathbb{R}$.

(v) $D_t^\gamma(rw) = rD_t^\gamma w + wD_t^\gamma r$.

(vi) $D_t^\gamma\left(\frac{w}{r}\right) = \frac{rD_t^\gamma w - wD_t^\gamma r}{r^2}, r \neq 0$.

(vii) $D_t^\gamma(w) = t^{1-\gamma}\frac{dw}{dt}$ when $w(t) = w$ is differentiable.

## 3 Governing equation

The BBM equation appeared in 1972 as an extension of the KdV model, which was developed to present surface water waves in a homogeneous system. An improvement of the KdV structure, the BBM equation finds applications beyond water surface waves, covering Rossby and drift waves in plasma. The BBM nonlinear system, as outlined in [38], is expressed as follows:

$$w_t + w_x + w^n w_x - w_{xxt} = 0, \tag{2}$$

and the KdV equation corresponds to the next form:

$$w_t + w_x + ww_x + w_{xxx} = 0. \tag{3}$$

The above Eqs (2) and (3) are fundamental instruments for understanding a diverse range of wave properties. They play crucial roles in the study of surface waves in water, acoustic and gravity waves in flexible fluids, hydromagnetic waves in plasmas, nonlinear dispersive processes' long waves, and harmonic crystals' acoustic waves, among other applications. In 2017, Wazwaz introduced a novel equation called the WBBM equation [39], which was derived from a modified three-dimensional BBM equation. This equation is expressed as follows:

$$w_t + w_x + w^2 w_y - w_{xzt} = 0, \tag{4}$$

$$w_t + w_z + w^2 w_x - w_{xyt} = 0, \tag{5}$$

$$w_t + w_y + w^2 w_z - w_{xxt} = 0. \tag{6}$$

This article will center on the previously mentioned equation Eq (5) from a fractional perspective, known as the second fractional 3D WBBM equation, as introduced in [31]. It is formulated as follows:

$$D_t^\gamma w + D_z^\gamma w + D_x^\gamma w^3 - D_{xyt}^{3\gamma} w = 0, \tag{7}$$

with real function $w(t, x, y, z)$ of free components $t$, $x$, $y$, and $z$. $D_t^\gamma, D_x^\gamma, D_y^\gamma$, and $D_z^\gamma$ implies the non-integer order $\gamma$ derivatives w.r. to $t$, $x$, $y$, and $z$, in sequence with $0 < \gamma \leq 1$, and $0 \leq t$. Applying the next traveling wave relation

$$w(x, y, z, t) = W(\varsigma), \varsigma = \frac{l_1}{\gamma}x^\gamma + \frac{l_2}{\gamma}y^\gamma + \frac{l_3}{\gamma}z^\gamma - \frac{l_4}{\gamma}t^\gamma, \tag{8}$$

on Eq (7) with $l_1 \neq 0, l_2 \neq 0, l_3 \neq 0$, and $l_4 \neq 0$, one reaches

$$(-l_4 + l_3)W' + l_1(W^3)' + l_1 l_2 l_4 W''' = 0. \tag{9}$$

Upon integrating Eq (9) w. r. to $\varsigma$, one arrive at

$$(-l_4 + l_3)W + l_1 W^3 + l_1 l_2 l_4 W'' + l_5 = 0, \tag{10}$$

where $l_5$ implies an integrating constant. For convenience, we set $l_5 = 0$, then Eq (10) turns

into the bellow-mentioned ordinary differential form

$$(-l_4 + l_3)W + l_1 W^3 + l_1 l_2 l_4 W'' = 0. \tag{11}$$

## 4 Bifurcation analysis

Bifurcation occurs in dynamic systems when small parameter changes cause qualitative changes in the system's behavior [40]. It often leads to the appearance of new stable states, periodic orbits, or chaotic behavior. Bifurcation theory helps to understand these sudden changes and predict system behavior in different states [41]. This paragraph offers an overview of the bifurcation and phase diagrams of the upcoming planner dynamic framework. By employing this methodology, qualitative analysis of nonlinear models becomes possible. There can be a range of trajectory shapes in this structure, including points, simple closed curves, or similar curves with varying shapes, representing solutions to Eq (7) in a variety of physical structures. Considering $\frac{dW}{d\varsigma} = P$, the planner dynamical framework for Eq (7) can be articulated as follows:

$$\frac{dW}{d\varsigma} = P, \frac{dP}{d\varsigma} = -\alpha W^3 - mW \tag{12}$$

for $\alpha = \frac{1}{l_2 l_4}$ and $m = \frac{l_3 - l_4}{l_1 l_2 l_4}$ with $l_3 \neq l_4$. By applying the first integral to Eq (12), one reaches the next Hamiltonian function

$$H(W, P) = \frac{1}{2}P^2 + \frac{\alpha}{4}W^4 + \frac{m}{2}W^2 = h, \tag{13}$$

which satisfies the Hamilton canonical equations $W' = \frac{\partial H}{\partial P}$ and $P' = -\frac{\partial H}{\partial W}$. Here $h$ is an integral constant known as the Hamiltonian constant or energy level. Additionally, it is also called the energy integral or total energy. On the other hand, $\frac{1}{2}P^2$ is the kinetic energy, and $\frac{\alpha}{4}W^4 + \frac{m}{2}W^2$ is the potential energy of the Hamiltonian system Eq (12).

Consider a real function $W(\varsigma)$, which is denoted as a solution to Eq (12) satisfying the physical constraints $\lim_{\varsigma \to -\infty} W(\varsigma) = a_1$ and $\lim_{\varsigma \to +\infty} W(\varsigma) = a_2$, with free constants $a_1$ and $a_2$. When $a_1 = a_2$, $W(\varsigma)$ represents a homoclinic trajectory, yielding $w(x, y, z, t)$ a solitary wave achievement for Eq (7). Moreover, if $a_1 \neq a_2$, $W(\varsigma)$ represents a heteroclinic orbit. When $a_1 > a_2$, $w(x, y, z, t)$ manifests as a kink wave outcome, whereas for $a_1 < a_2$, it becomes an anti-kink wave result for Eq (7). A closed-phase portrait is displayed by Eq (12) in another scenario, which results in a periodic solution for Eq (7). It is worth noting that a phase portrait describes an orbit collection inside a phase plane.

The structure Eq (12)'s equilibrium points are determined by solving the resulting set $P = 0$, $-\alpha W^3 - mW = 0$. When $\alpha m > 0$, only one equilibrium point $(0, 0)$ is identified. However, when $\alpha m < 0$, three equilibrium points are obtained, namely $(0, 0)$, $\left(\sqrt{-\frac{m}{\alpha}}, 0\right)$, and $\left(-\sqrt{-\frac{m}{\alpha}}, 0\right)$.

The Jacobian matrix of structure Eq (12) takes the next determinant formation:

$$D(W, P) = \begin{vmatrix} 0 & 0 \\ -3\alpha W^2 - m & 0 \end{vmatrix} = 3\alpha W^2 + m. \tag{14}$$

Hence, the characteristic value of Eq (12) at $(W, 0)$ is given by $\sqrt{-3\alpha W^2 - m}$. As a result, the equilibrium point $(W, 0)$ is signified as a central equilibrium for $D(W, P) > 0$, a saddle

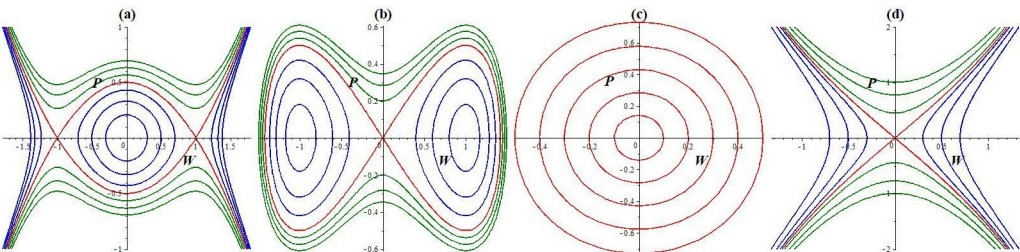

**Fig 1. Phase portraits within the dynamic framework Eq (12).** (a) For $\alpha < 0, m > 0$, (b) $\alpha > 0, m < 0$, (c) $\alpha, m > 0$, and (d) $\alpha, m < 0$.

point for $D(W, P) < 0$, and a cuspidal point for $D(W, P) = 0$. Various parameters can produce the forthcoming possible conditions (see Fig 1a–1d):

**Case 1:** $\alpha < 0, m > 0$

Three equilibrium points, $(0, 0)$, $(1, 0)$, and $(-1, 0)$, can be obtained through imposing parameter settings as follows: $l_4 = 2, l_1 = l_3 = 1$, and $l_2 = -1$; these are shown in Fig 1a. It is obvious from the figure that $(0, 0)$ denotes a central equilibrium, whereas both $(1, 0)$ and $(-1, 0)$ signify saddle points. Fig 1a visibly exhibits the presence of anti-kink and kink wave properties, which are facilitated by the linking of two heteroclinic trajectories $(-1, 0)$ and $(1, 0)$.

**Case 2:** $\alpha > 0, m < 0$

Three equilibrium points, $(0, 0)$, $(1, 0)$, and $(-1, 0)$, can be obtained through imposing parameter settings as follows: $l_1 = l_2 = l_3 = 1$, and $l_2 = 2$; these are pictured in Fig 1b. It is obvious from the figure that $(0, 0)$ denotes a saddle point, whereas both $(1, 0)$ and $(-1, 0)$ signify centrs. The paths consist of closed curves that include a variety of outcomes, including two homoclinic orbits (which are red) through $(0, 0)$, two families of periodic orbits (which are blue) around $(1, 0)$ and $(-1, 0)$, and a family of hyperperiodic orbits (which are green).

**Case 3:** $\alpha > 0, m > 0$

Only one equilibrium point, $(0, 0)$, can be obtained by imposing parameter settings as follows: $l_1 = l_3 = 1$, and $l_2 = l_4 = -1$, which is presented in Fig 1c. It is obvious from the figure that $(0, 0)$ denotes a central equilibrium. The paths consist of closed curves that contain a family of periodic orbits (which are red) around $(0, 0)$. As a result, system Eq (12) contains a periodic wave solution.

**Case 4:** $\alpha < 0, m < 0$

Only one equilibrium point, $(0, 0)$, can be obtained by imposing parameter settings as follows: $l_1 = l_2 = l_3 = 1$, and $l_4 = -1$, which is presented in Fig 1d. It is obvious from the figure that $(0, 0)$ denotes a saddle point. In this case, the trajectories do not encompass closed orbits for the structure Eq (12).

## 5 Chaotic behaviors

Chaos refers to a type of behavior exhibited by specific dynamic systems that may appear random but are influenced by deterministic rules [42]. Chaotic systems are highly sensitive to

initial conditions, meaning even minor alterations in initial values can result in significantly divergent outcomes over time. Although such systems often exhibit intricate and unpredictable behavior, they can also display underlying shapes or structures. Chaotic behavior can occur in a variety of natural and artificial systems, including weather patterns, optics, and fluid dynamics [43, 44]. This section investigates the chaotic characteristics of the resulting dynamic structure by evaluating the disturbed form. This exploration involves studying 3D and 2D phase views. To begin this examination, here we assume $\frac{dW}{d\varsigma} = P$, then the dynamic system is:

$$\frac{dW}{d\varsigma} = P, \frac{dP}{d\varsigma} = -\alpha W^3 - mW + \sigma\cos(\omega\varsigma), \tag{15}$$

Here, $\sigma\cos(\omega t)$ denotes the disturbed form, where $\sigma$ represents the intensity and $\omega$ denotes the system's frequency. In this paragraph, we investigate how the strength and frequency of the allotment affect the structure described in Eq (15). While keeping the key parameters unchanged ($l_1 = l_2 = 1, l_3 = -\frac{2}{3}, l_4 = \frac{1}{3}$), we observe chaotic and quasi-periodic patterns with various frequencies and intensities, as depicted in Figs 2a–2c, 3a–3c and 4a–4c. Fig 2a–2c represents the status of Eq (15) with $\sigma = 0$. We depict the system's trajectory position following the allocation's intensity and frequency. As shown in Fig 2, Eq (15) exhibits quasi-periodic behavior in time series plots and 2D and 3D phase diagrams. As shown in Fig 3a–3c, the dynamic structure transitions from a quasi-periodic to a chaotic position with a slight increase in intensity and frequency ($\sigma$ rises to 0.3 and $\omega = 2.2$). Moreover, the system experiences a chaotic state even for significant disturbances in frequency and intensity ($\sigma$ rises to 1.4 and $\omega = 3.9$) (see Fig 4a–4c). The multistability of Eq (15) is shown in Fig 5a–5c for $l_1 = l_2 = 1, l_3 = -\frac{2}{3}, l_4 = \frac{1}{3}, \sigma = 1.9$, and $\omega = 3.9$ with different initial settings. We observe the quasi-periodic nature

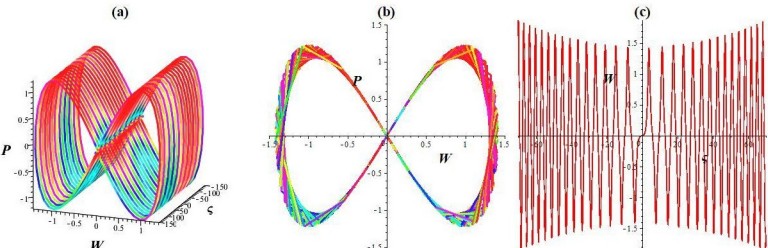

**Fig 2. Quasi-periodic structure within the framework Eq (15) for initial setting (0, 0.01).** (a) 3D phase diagram, (b) 2D phase diagram, and (c) Time series plot.

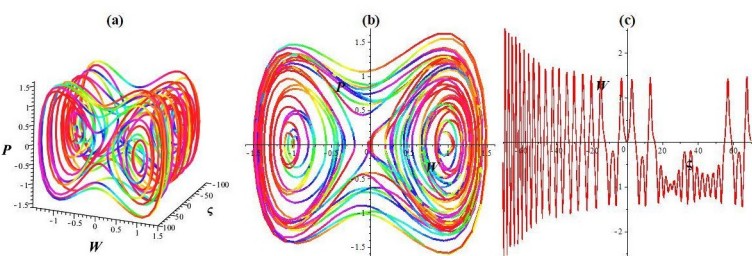

**Fig 3. Chaotic nature within the framework Eq (15) for initial setting (0, 0.01).** (a) 3D phase diagram, (b) 2D phase diagram, and (c) Time series plot.

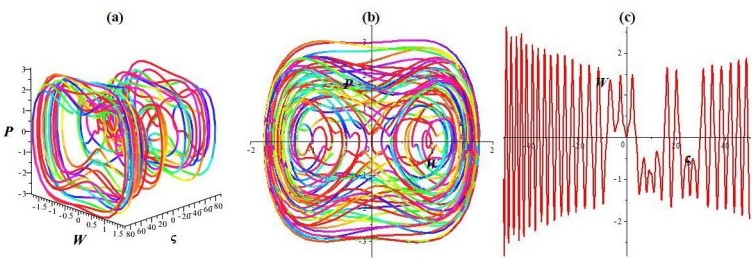

**Fig 4. Chaotic nature within the framework Eq (15) for initial setting (0, 0.01).** (a) 3D phase diagram, (b) 2D phase diagram, and (c) Time series plot.

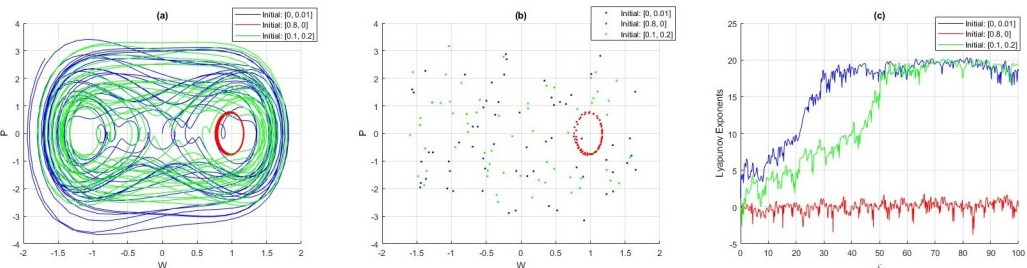

**Fig 5. Multistability within the framework Eq (15).** (a) 2D phase diagram, (b) Poincaré plot, and (c) Lyapunov plot.

of Eq (15) for initial value (0.8, 0) and chaotic nature for initial values (0, 0.01) and (0.1, 0.2). Therefore the chaotic behavior of Eq (15) is observed under external disturbances with different initial values.

## 6 Sensitivity analysis

Sensitivity analysis assesses how different sources of uncertainty in an input can impact a mathematical model's output [45]. It helps to identify which inputs have the most significant impact on output and how variations in inputs are transmitted through the system. Sensitivity analysis helps to understand the strength, reliability, and credibility of model predictions that may guide decision-making in complex systems [46]. This segment explores how initial conditions affect the disturbed form described by Eq (15) across various intensities and frequencies while keeping the parameter values unchanged ($l_1 = l_2 = 1, l_3 = -\frac{2}{3}, l_4 = \frac{1}{3}$). The resulting time series images for the initial settings (0, 0.01) and (0, 0.02) are depicted in Fig 6a–6c by the

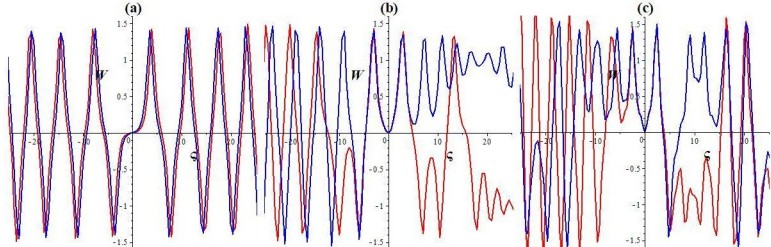

**Fig 6. Sensitive nature within the framework Eq (15).** (a) For $\sigma = 0$, (b) for $\sigma = 0.3$, $\omega = 2.2$, and (c) for $\sigma = 1.4$, $\omega = 3.9$.

red and blue curves, respectively. Fig 6a illustrates that the periodic behavior of the result depends on the initial setting of the disturbed structure for the absence of distribution intensity ($\sigma = 0$). Fig 6b shows that when the distribution intensity is low ($\sigma = 0.3$), the blue and red time series designs show only minor alterations, signifying a low sensitivity to the initial state. Additionally, as a result of the improvement in the intensity of the perturbation ($\sigma = 1.4$), as shown in Fig 6c, significant discrepancies are observed between the time series plots, suggesting high sensitivity to variations in the initial conditions.

## 7 Soliton waveforms of the adopted nonlinear equation

The purpose of this segment is to scrutinize diverse solitons attained from the stated nonlinear framework using the planner dynamics procedure.

**Case 1:** $\alpha < 0$, and $m > 0$

For $h \in \left(0, -\frac{m^2}{4\alpha}\right)$, a group of periodic trajectories for the dynamic structure expressed in Eq (12) can be obtained. For this case, the Hamiltonian function takes on the following structure:

$$P = \pm\sqrt{-\frac{\alpha}{2}}\sqrt{W^4 + \frac{2m}{\alpha}W^2 - \frac{4h}{\alpha}} = \pm\sqrt{-\frac{\alpha}{2}}\sqrt{(S_1^2 - W^2)(S_2^2 - W^2)}, \tag{16}$$

with $S_1 = \sqrt{-\frac{m}{\alpha} + \frac{\sqrt{m^2 + 4\alpha h}}{\alpha}}$ and $S_2 = \sqrt{-\frac{m}{\alpha} - \frac{\sqrt{m^2 + 4\alpha h}}{\alpha}}$. Substituting Eq (16) in the initial part of the Hamiltonian framework Eq (12) and then integrating yields:

$$\int_0^Q \frac{dR}{\sqrt{(S_1^2 - R^2)(S_2^2 - R^2)}} = \pm\sqrt{-\frac{\alpha}{2}}(\varsigma - \varsigma_0), \tag{17}$$

where the integration constant is denoted by $\varsigma_0$. Hence, we arrive at the following pair of periodic wave results:

$$w_1(x, t) = \pm S_1 sn(S_2\sqrt{-\frac{\alpha}{2}}(\frac{l_1}{\gamma}x^\gamma + \frac{l_2}{\gamma}y^\gamma + \frac{l_3}{\gamma}z^\gamma - \frac{l_4}{\gamma}t^\gamma - \varsigma_0)\frac{S_1}{S_2}).$$

When we set $h = -\frac{m^2}{4\alpha}$, the condition $S_1^2 = S_2^2 = -\frac{m}{\alpha}$ holds, resulting in the subsequent kink wave outcome if we consider the positive expression and antikink wave outcome if we consider the negative expression:

$$w_2(x, t) = \pm\sqrt{-\frac{m}{\alpha}}\tanh(\sqrt{\frac{m}{2}}(\frac{l_1}{\gamma}x^\gamma + \frac{l_2}{\gamma}y^\gamma + \frac{l_3}{\gamma}z^\gamma - \frac{l_4}{\gamma}t^\gamma - \varsigma_0)).$$

**Case 2:** $\alpha > 0$, and $m < 0$

For $h \in \left(-\frac{m^2}{4\alpha}, 0\right)$, we achieve 2 groups of periodic trajectories of the dynamical system Eq (12). For this condition, the Hamiltonian structure can be expressed in the upcoming form

$$P = \pm\sqrt{\frac{\alpha}{2}}\sqrt{-W^4 - \frac{2m}{\alpha}W^2 + \frac{4h}{\alpha}} = \pm\sqrt{\frac{\alpha}{2}}\sqrt{(W^2 - S_1^2)(S_2^2 - W^2)}, \tag{18}$$

whereas $S_1 = \sqrt{-\frac{m}{\alpha} + \frac{\sqrt{m^2 + 4\alpha h}}{\alpha}}$ and $S_2 = \sqrt{-\frac{m}{\alpha} - \frac{\sqrt{m^2 + 4lh}}{\alpha}}$. Substituting Eq (18) in the initial part of the Hamiltonian framework Eq (12) and then integrating yields:

$$\int_W^{S_2} \frac{dR}{\sqrt{(R^2 - S_1^2)(S_2^2 - R^2)}} = \mp\sqrt{\frac{\alpha}{2}}(\varsigma - \varsigma_0), \tag{19}$$

and

$$\int_{-S_2}^{W} \frac{dR}{\sqrt{(R^2 - S_1^2)(S_2^2 - R^2)}} = \pm\sqrt{\frac{\alpha}{2}}(\varsigma - \varsigma_0), \tag{20}$$

where the integration constant is denoted by $\varsigma_0$. Hence, we arrive at the following pair of periodic wave results:

$$w_3(x,t) = \pm S_1 dn(S_1\sqrt{\frac{\alpha}{2}}(\frac{l_1}{\gamma}x^\gamma + \frac{l_2}{\gamma}y^\gamma + \frac{l_3}{\gamma}z^\gamma - \frac{l_4}{\gamma}t^\gamma - \varsigma_0), \frac{\sqrt{S_1^2 - S_2^2}}{S_1}).$$

If $h = 0$, we find that $S_1 = \sqrt{-\frac{2m}{\alpha}}$ and $S_2 = 0$. This results in the emergence of the subsequent two solitary wave results: dark soliton if we consider the negative expression and bright soliton if we take into account the positive expression.

$$w_4(x,t) = \pm\sqrt{-\frac{2m}{\alpha}}\mathrm{sech}(\sqrt{-m}(\frac{l_1}{\gamma}x^\gamma + \frac{l_2}{\gamma}y^\gamma + \frac{l_3}{\gamma}z^\gamma - \frac{l_4}{\gamma}t^\gamma - \varsigma_0)).$$

If $0 < h < +\infty$, then the Hamiltonian function takes on the next structure:

$$P = \pm\sqrt{\frac{\alpha}{2}}\sqrt{-W^4 - \frac{2m}{\alpha}W^2 + \frac{4h}{\alpha}} = \pm\sqrt{\frac{\alpha}{2}}\sqrt{(S_1^2 - W^2)(S_3^2 + W^2)}, \tag{21}$$

whereas $S_1 = \sqrt{-\frac{m}{\alpha} + \frac{\sqrt{m^2 + 4\alpha h}}{\alpha}}$ and $S_3 = \sqrt{\frac{m}{\alpha} + \frac{\sqrt{m^2 + 4\alpha h}}{\alpha}}$. Substituting Eq (21) in the initial part of the Hamiltonian framework Eq (12) and then integrating yields:

$$\int_0^{W} \frac{dR}{\sqrt{(S_1^2 - R^2)(S_3^2 + R^2)}} = \pm\sqrt{\frac{\alpha}{2}}(\varsigma - \varsigma_0), \tag{22}$$

where the integration constant is denoted by $\varsigma_0$. Hence, we arrive at the following pair of periodic wave results:

$$w_5(x,t) = \pm S_1 cn(\sqrt{\frac{\alpha(S_1^2 + S_3^2)}{2}}(\frac{l_1}{\gamma}x^\gamma + \frac{l_2}{\gamma}y^\gamma + \frac{l_3}{\gamma}z^\gamma - \frac{l_4}{\gamma}t^\gamma - \varsigma_0), \frac{S_1}{\sqrt{S_1^2 + S_3^2}}).$$

## 8 Graphical representations

Using appropriate parameter selection, this section explores graphical representations of the acquired results, followed by an explanation of their physical significance. Solution $w_4$ manifests both bright and dark solitons. To visually present the physical behavior of the specific result $w_4$ under $\varsigma_0 = 1, \gamma = 0.5, l_1 = l_2 = 1, l_3 = \frac{1}{2}, l_4 = 2$, and $h = 0$ at $y = 1, z = 1$, a numerical illustration is depicted in Fig 7a–7d. As shown in Fig 7a and 7b, a bright waveform is obvious for positive values, while Fig 7c and 7d demonstrate a dark waveform for negative values. Another nature demonstrated in Fig 8a–8d clarifies the physical features of the outcome $w_2$ under $\varsigma_0 = 1, \gamma = 0.5, l_1 = 2, l_2 = l_3 = -1, l_4 = 1$, and $h = \frac{1}{4}$ at $y = 1, z = 1$. Fig 8a and 8b showcase a kink waveform for positive values, while Fig 8c and 8d express an anti-kink soliton for negative values. Outcomes $w_1$, $w_3$, and $w_5$ display a periodic waveform. Additionally, to elucidate the physical attributes of the specific solution $w_1$ under $\varsigma_0 = 1, \gamma = 0.8, l_1 = 2, l_2 = l_3 = -1, l_4 = 1$, and $h = \frac{1}{5}$ at $y = 1, z = 1$, a graphical representation is offered in Fig 9a and 9b.

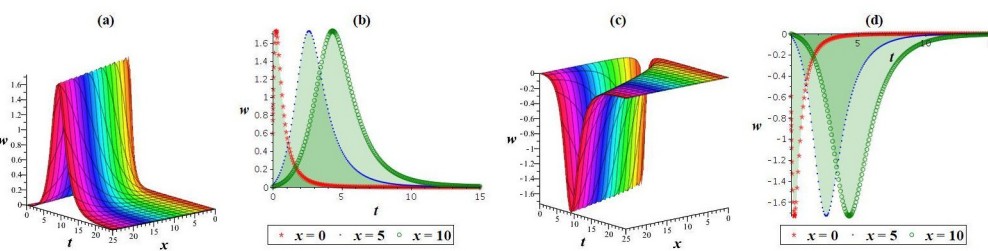

**Fig 7. Graphics of outcome $w_4$: (a,c) cubic plot, (b,d) 2D shape.**

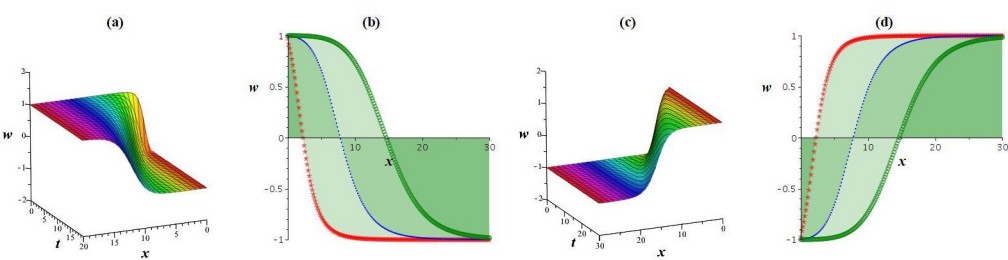

**Fig 8. Waveform of solution $w_2$: (a,c) cubic plot, (b,d) 2D shape.**

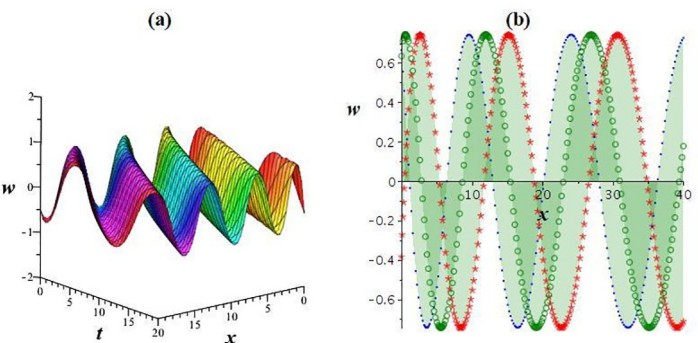

**Fig 9. View of periodic wave solution $w_1$.** (a) cubic plot, (b) 2D plot.

## 9 Stability assessment

To examine the stability of the governing model, the current document will perform the linear stability concept, outlined in a source cited as [47]. We will assume the mentioned system's integer order, as specified in equation Eq (4). Subsequently, the perturbed solution will take the below-mentioned form:

$$w(t, x, y, z) = a + v\Phi(t, x, y, z), \tag{23}$$

whilst $a$ corresponds to a stable-state solution of Eq (4). Combining Eqs (23) and (4), we

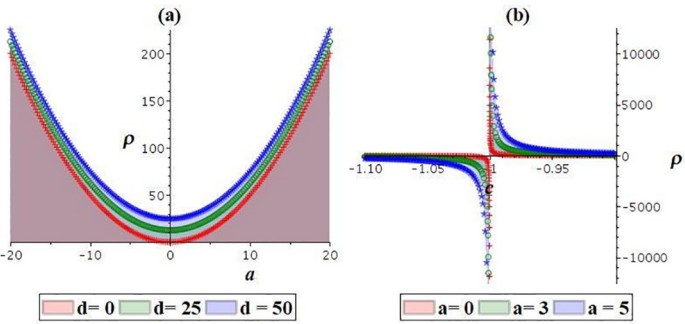

**Fig 10. Stability plot of Eq (28): (a) stable propagation for $b = c = 1$, (b) unstable propagation for $b = 1$, $d = 1$.**

achieve

$$v(\Phi_x\Phi^2v^2 + 2\Phi_x\Phi va + \Phi_x a^2 + \Phi_t + \Phi_z - \Phi_{xyt}) = 0, \tag{24}$$

Linearizing the current equation as expressed in the form of $v$ gives,

$$\Phi_x a^2 + \Phi_t + \Phi_z - \Phi_{xyt} = 0, \tag{25}$$

Now, let us presume that the subsequent solution to the present framework

$$\Phi(t, x, y, z) = e^{i(bx+cy+dz-\rho t)}, \tag{26}$$

whilst $b$, $c$, $d$ correspond to normalized wave number and $\rho$ corresponds to the frequency of perturbation. Using Eqs (26) and (25) yields

$$\rho bc - ba^2 + \rho - d = 0, \tag{27}$$

Solving the present equation Eq (27) for $\rho$ gives

$$\rho(b, c, d) = \frac{d + ba^2}{bc + 1}, bc \neq -1. \tag{28}$$

The stability behavior of equation Eq (28) is plotted in Fig 10a and 10b. It can be observed that equation Eq (28) remains stable propagation for $b = 1$, $c = 1$, since for the selected parameters, the figure displays a continuous and smooth function $\rho(b, c, d)$ (refer to Fig 10a). On the other hand, the function $\rho(b, c, d)$ remains unstable propagation for $b = 1$, $d = 1$, since for the selected parameters, the denominator $bc + 1$ approaches zero when $c$ is chosen close to $-1$ (refer to Fig 10b).

## 10 Novelty of the outcomes

This paragraph aims to illustrate the uniqueness of our achievements by comparing them with those of previous works. By systematically comparing our results with those in the literature [31, 48–50], we highlight the novelty of our results and the uniqueness of our implemented technique. Seadawy and his coauthors obtained periodic and hyperbolic function outcomes to the governing nonlinear problem by employing the simple ansatz technique [48]. Mamun et al. extracted trigonometric and hyperbolic function solutions of the mentioned model through the $(G'/G^2)$-expansion technique [31]. Demirbilek derived exponential function solutions for the mentioned model by employing the IBSEF technique [49]. Inc and his

collaborators presented generalized trigonometric and hyperbolic function solutions to this model through the Sarder-subequation scheme [50]. In contrast to these existing procedures and outcomes, our results, denoted as $q_1$, $q_2$, $q_3$, $q_4$, and $q_5$, diverge significantly, highlighting their novelty. Specifically, we present analyses such as bifurcation, chaos, stability, and sensitivity analysis that were previously ignored. As a result of this study, we provide novel insights into the dynamics and behavior of the model, representing the originality of our contribution to the field.

## 11 Conclusion

The bifurcation, chaos, and stability analysis for the second fractional 3D WBBM equation has provided significant insights into shallow water wave dynamics. By applying the Galilean transformation, we achieve a dynamic structure that facilitates a comprehensive analysis of bifurcations. Furthermore, our exploration of various solitary wave findings, encompassing periodic waves, bright solitons, dark solitons, anti-kink waves, and kink waves, provided a nuanced understanding of their properties and existence, as visually showcased through simulations.

As shown in our findings, the integration methods employed throughout our study are effective, concise, and efficient. Our findings not only advance our understanding of nonlinear phenomena in shallow water waves but also suggest the possibility of applying our methodologies to more complex nonlinear systems encountered in contemporary engineering scientific contexts. Overall, this study provides avenues for further exploration and emphasizes the significance of interdisciplinary approaches for addressing complex challenges in modern research.

## Author Contributions

**Conceptualization:** M. Zulfikar Ali.

**Methodology:** Mohammad Safi Ullah.

**Software:** Mohammad Safi Ullah.

**Supervision:** M. Zulfikar Ali, Harun-Or Roshid.

**Validation:** M. Zulfikar Ali, Harun-Or Roshid.

**Writing – original draft:** Mohammad Safi Ullah.

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
