## [Decision Letter · Decision Letter 0]

19 Jun 2024

PONE-D-24-22626Bifurcation, chaos, and stability analysis  to the second fractional WBBM modelPLOS ONE

Dear Dr. Ullah,

Thank you for submitting your manuscript to PLOS ONE. Reviews strongly suggest that a major revision is necessary. In particular, one review, presented by one of top experts in the field, recommends outright rejection.  You have an option to thoroughly revise the paper and resubmit it.

We look forward to receiving your revised manuscript.

Best regards,

Boris Malomed

Academic Editor

PLOS ONE

Journal Requirements:

Reviewers' comments:

Reviewer's Responses to Questions

**Comments to the Author**

1. Is the manuscript technically sound, and do the data support the conclusions?

Reviewer #1: Yes

Reviewer #2: Yes

Reviewer #3: Yes

Reviewer #4: Yes

2. Has the statistical analysis been performed appropriately and rigorously? 

Reviewer #1: N/A

Reviewer #2: N/A

Reviewer #3: Yes

Reviewer #4: Yes

3. Have the authors made all data underlying the findings in their manuscript fully available?

Reviewer #1: Yes

Reviewer #2: Yes

Reviewer #3: Yes

Reviewer #4: Yes

4. Is the manuscript presented in an intelligible fashion and written in standard English?

Reviewer #1: Yes

Reviewer #2: Yes

Reviewer #3: Yes

Reviewer #4: No

5. Review Comments to the Author

Reviewer #1: DESCRIPTION OF THIS ARTICLE:

In this article the authors study a fractional generalization of the WBBM (Wazwaz-Benjamin-Bona-Mahony) model. The fractional derivatives used are conformable derivatives. By proposing an adequate similarity reduction [Eq. (8)], the authors managed to transform the partial differential equation Eq. (7) into the ordinary differential equation (ODE) (9), which can be integrated once, to yield the second-order Eq. (10). This transformation is, by itself, an excellent result. Then Eq. (10) is transformed into a system of two first-order ODEs [Eqs. (12)] which depend on two parameters: “l” (afterwards called “alfa”] and “m”. From system (12) the phase portraits shown in Fig. 1 are constructed, showing three or one equilibrium points (depending on the signs of the parameters “alfa” and “m”). Then the authors show that system (12) has quasi-periodic, periodic, and chaotic solutions, which is an interesting result. The apparition of chaotic solutions is then studied by introducing a time-dependent oscillatory perturbation [shown in Eqs. (15)]. Moreover, the sensitivity of the solutions to small changes in the initial conditions [Fig. 6] is also studied. Finally, exact analytical solutions are obtained. Periodic solution and soliton-like solutions are obtained.

OPINION OF THE ARTICLE:

This is an interesting article. It is clear and well-written. I consider that it deserves to be published in PLOS ONE in its present form. Only one small detail has to be corrected. In the system (12) two coefficients appear: “l” and “m”. However, in the following, the parameter “l” is called “alpha”. Therefore, I believe that Eqs. (12) and (13), and the paragraph below Eq. (13), should be corrected, writing “alpha” instead of “l”.

RECOMMENDATION:

I consider that this article is suitable for publication in PLOS ONE (once the small error mentioned above is corrected).

Reviewer #2: See attachment.

Manuscript Number: PONE-D-24-22626

Full Title: Bifurcation, chaos, and stability analysis to the second fractional WBBM model

By Mohammad Safi Ullaha, M. Zulfikar Ali, Harun-Or Roshid

In this manuscript, the authors presented the bifurcation, chaos, and stability analysis for a significant the second 3D fractional Wazwaz-Benjamin-Bona-Mahony (WBBM) model in the research of shallow water waves. In addition to improving our understanding of shallow water nonlinear dynamics, including waveform features, bifurcation analysis, sensitivity, and stability, this study reveals insights into dynamic properties and wave patterns. The obtained results are interesting and novel. However, the following issues must be modified before it is considered for publication.

1. In Eq. (11), it must be emphasized that . Otherwise, the bifurcation analysis mentioned below cannot occur.

2. For the literature on the Nizhnik Novikov Veselov model and various others, please introduce the latest achievements: Chaos, Solitons and Fractals 176 (2023) 114075 and Chinese Physics Letters 41, 044201 (2024).

3. In Eq. (8), to prevent singularities in future analysis, it is necessary to propose limiting conditions , , , .

4. On the next line of Eq. (12), it is mentioned that , but Eq. (12) does not have this symbol at all. Please carefully check.

5. If Eq. (12) is , the label in Fig. 1 must be modified, from change to .

6. After Eq. (12), all symbols must be carefully checked to prevent mixing of and .

In conclusion, if the author can make revisions according to the above suggestions, I recommend publishing this article in PLOS ONE.

Reviewer #3: The paper investigated theoretically and numerically the bifurcation, chaos, and stability analysis to the second 3D fractional WBBM model, stressing the findings of some solitary waves as bright solitons, dark solitons, kink waves, and anti-kink waves. The novelty of the results was justified. This work can be recommended to accept in PLOS one, while before that a critical issue should be well taken into consideration.

Such issue is the stability of the diverse soliton structures predicted in the manuscript, although the stability assessment was done in the Section 9, it is instructive to include at least one stable example and an unstable one in the propagation course.

Reviewer #4: In this manuscript “Bifurcation, chaos, and stability analysis to the second fractional WBBM model”, authors had investigated the Bifurcation, chaos, and solitons in the fractional 3D WBBM equation.

The issues studied in this paper have been analyzed and presented in other papers, such as:

1.“Periodic and solitary wave solutions to a family of new 3D fractional WBBM equations using the two-variable method” published in “Partial Differential Equations in Applied Mathematics 3 (2021) 100033”.

2.“Bifurcation analysis and new waveforms to the first fractional WBBM equation”, published in “Scientifc Reports (2024) 14:11907”.

The theoretical results presented in this paper have been does not introduce new concepts or significant advancements in the field. The issues and model discussed in this paper are similar to the previous work of “Scientifc Reports (2024) 14:11907”, however, the authors did not provide a detailed comparison with research.

After a thorough review of the manuscript, I regret to inform you that I cannot recommend its publication.

6. PLOS authors have the option to publish the peer review history of their article (what does this mean?). If published, this will include your full peer review and any attached files.

Reviewer #1: No

Reviewer #2: No

Reviewer #3: **Yes: **Jianhua Zeng

Reviewer #4: No

---

## [Author Response · Author response to Decision Letter 0]

27 Jun 2024

Response To Reviewer’s Comments

Reviewer #1: 

DESCRIPTION OF THIS ARTICLE:

In this article the authors study a fractional generalization of the WBBM (Wazwaz-Benjamin-Bona-Mahony) model. The fractional derivatives used are conformable derivatives. By proposing an adequate similarity reduction [Eq. (8)], the authors managed to transform the partial differential equation Eq. (7) into the ordinary differential equation (ODE) (9), which can be integrated once, to yield the second-order Eq. (10). This transformation is, by itself, an excellent result. Then Eq. (10) is transformed into a system of two first-order ODEs [Eqs. (12)] which depend on two parameters: “l” (afterwards called “alfa”] and “m”. From system (12) the phase portraits shown in Fig. 1 are constructed, showing three or one equilibrium points (depending on the signs of the parameters “alfa” and “m”). Then the authors show that system (12) has quasi-periodic, periodic, and chaotic solutions, which is an interesting result. The apparition of chaotic solutions is then studied by introducing a time-dependent oscillatory perturbation [shown in Eqs. (15)]. Moreover, the sensitivity of the solutions to small changes in the initial conditions [Fig. 6] is also studied. Finally, exact analytical solutions are obtained. Periodic solution and soliton-like solutions are obtained.

Response: Thanks for your real view and constructive comments.

OPINION OF THE ARTICLE:

This is an interesting article. It is clear and well-written. I consider that it deserves to be published in PLOS ONE in its present form. Only one small detail has to be corrected. In the system (12) two coefficients appear: “l” and “m”. However, in the following, the parameter “l” is called “alpha”. Therefore, I believe that Eqs. (12) and (13), and the paragraph below Eq. (13), should be corrected, writing “alpha” instead of “l”.

Response: Sorry for the typing mistake. We remove this problem.

RECOMMENDATION:

I consider that this article is suitable for publication in PLOS ONE (once the small error mentioned above is corrected).

Response: We appreciate your help in improving our article.

Reviewer #2: 

See attachment.

Manuscript Number: PONE-D-24-22626

Full Title: Bifurcation, chaos, and stability analysis to the second fractional WBBM model

By Mohammad Safi Ullah, M. Zulfikar Ali, Harun-Or Roshid

In this manuscript, the authors presented the bifurcation, chaos, and stability analysis for a significant the second 3D fractional Wazwaz-Benjamin-Bona-Mahony (WBBM) model in the research of shallow water waves. In addition to improving our understanding of shallow water nonlinear dynamics, including waveform features, bifurcation analysis, sensitivity, and stability, this study reveals insights into dynamic properties and wave patterns. The obtained results are interesting and novel. However, the following issues must be modified before it is considered for publication.

 In Eq. (11), it must be emphasized that . Otherwise, the bifurcation analysis mentioned below cannot occur.

Response: We mark the constraint condition after Eq. (11). 

2. For the literature on the Nizhnik Novikov Veselov model and various others, please introduce the latest achievements: Chaos, Solitons and Fractals 176 (2023) 114075 and Chinese Physics Letters 41, 044201 (2024).

Response: We included some relevant research papers in the reference part and cited them in the text. 

3. In Eq. (8), to prevent singularities in future analysis, it is necessary to propose limiting conditions , , , .

Response: Thanks for your valuable suggestions. We propose these limiting conditions.

4. On the next line of Eq. (12), it is mentioned that , but Eq. (12) does not have this symbol at all. Please carefully check.

Response: Sorry for the typing mistake. We remove this problem.

5. If Eq. (12) is , the label in Fig. 1 must be modified, from change to .

Response: Sorry for the typing mistake. We remove this problem.

6. After Eq. (12), all symbols must be carefully checked to prevent mixing of and .

Response: Sorry for the typing mistake. We remove this problem.

In conclusion, if the author can make revisions according to the above suggestions, I recommend publishing this article in PLOS ONE.

Response: Thanks for helping us to improve our article.

Reviewer #3:

The paper investigated theoretically and numerically the bifurcation, chaos, and stability analysis to the second 3D fractional WBBM model, stressing the findings of some solitary waves as bright solitons, dark solitons, kink waves, and anti-kink waves. The novelty of the results was justified. This work can be recommended to accept in PLOS one, while before that a critical issue should be well taken into consideration.

Response: Thanks for your real view and constructive comments.

Such issue is the stability of the diverse soliton structures predicted in the manuscript, although the stability assessment was done in the Section 9, it is instructive to include at least one stable example and an unstable one in the propagation course.

Response: Thanks, we include both stable and unstable examples in the propagation course. Please accept our sincere thanks for your suggestions. 

Reviewer #4: 

In this manuscript “Bifurcation, chaos, and stability analysis to the second fractional WBBM model”, authors had investigated the Bifurcation, chaos, and solitons in the fractional 3D WBBM equation.

The issues studied in this paper have been analyzed and presented in other papers, such as:

1.“Periodic and solitary wave solutions to a family of new 3D fractional WBBM equations using the two-variable method” published in “Partial Differential Equations in Applied Mathematics 3 (2021) 100033”.

Response: Dear respected reviewer, thanks for your valuable comments. In my observation, the 3D fractional WBBM model contains the following three equations.

D■(γ@t)w+D■(γ@x)w+D■(γ@y)w^3-D■(3γ@xzt)w=0 (1)

D■(γ@t)w+D■(γ@z)w+D■(γ@x)w^3-D■(3γ@xyt)w=0 (2)

D■(γ@t)w+D■(γ@y)w+D■(γ@z)w^3-D■(3γ@xxt)w=0 (3)

The reviewers mentioned article Our article

The reviewers mentioned article describes the following first fractional model 

 D■(γ@t)w+D■(γ@x)w+D■(γ@y)w^3-D■(3γ@xzt)w=0 (1) Our article describes the following second fractional model

D■(γ@t)w+D■(γ@z)w+D■(γ@x)w^3-D■(3γ@xyt)w=0 (2)

The reviewers mentioned article describes equation (1) by (G'/G,1/G) expansion process.

Our article describes equation (2) with Bifurcation, chaos, sensitivity analysis, stability analysis, and multistability analysis.

2.“Bifurcation analysis and new waveforms to the first fractional WBBM equation”, published in “Scientifc Reports (2024) 14:11907”.

The theoretical results presented in this paper have been does not introduce new concepts or significant advancements in the field. The issues and model discussed in this paper are similar to the previous work of “Scientifc Reports (2024) 14:11907”, however, the authors did not provide a detailed comparison with research.

Response: The 3D fractional WBBM model contains the previously mentioned three equations and the following differences are observed.

The reviewers mentioned article Our article

The reviewers mentioned article describes the following first fractional model 

 D■(γ@t)w+D■(γ@x)w+D■(γ@y)w^3-D■(3γ@xzt)w=0 (1) Our article describes the following second fractional model

D■(γ@t)w+D■(γ@z)w+D■(γ@x)w^3-D■(3γ@xyt)w=0 (2)

The reviewer’s mentioned article does not contain the Lyapunov exponent and multistability analysis. Our article contains both the Lyapunov exponent and multistability analysis.

After a thorough review of the manuscript, I regret to inform you that I cannot recommend its publication.

Response: The reviewer’s mentioned articles contain the first fractional model but our investigation contains the second fractional model. Please recheck and reconsider our work. We appreciate your assistance in improving our article.

---

## [Decision Letter · Decision Letter 1]

9 Jul 2024

Bifurcation, chaos, and stability analysis  to the second fractional WBBM model

PONE-D-24-22626R1

Dear Dr. Ullah,

We’re pleased to inform you that your manuscript has been judged scientifically suitable for publication and will be formally accepted for publication once it meets all outstanding technical requirements.

Kind regards,

Boris Malomed

Academic Editor

PLOS ONE

Additional Editor Comments (optional):

Reviewers' comments:

Reviewer's Responses to Questions

**Comments to the Author**

1. If the authors have adequately addressed your comments raised in a previous round of review and you feel that this manuscript is now acceptable for publication, you may indicate that here to bypass the “Comments to the Author” section, enter your conflict of interest statement in the “Confidential to Editor” section, and submit your "Accept" recommendation.

Reviewer #1: All comments have been addressed

Reviewer #2: All comments have been addressed

Reviewer #4: All comments have been addressed

2. Is the manuscript technically sound, and do the data support the conclusions?

Reviewer #1: Yes

Reviewer #2: Yes

Reviewer #4: Yes

3. Has the statistical analysis been performed appropriately and rigorously? 

Reviewer #1: N/A

Reviewer #2: Yes

Reviewer #4: Yes

4. Have the authors made all data underlying the findings in their manuscript fully available?

Reviewer #1: Yes

Reviewer #2: No

Reviewer #4: Yes

5. Is the manuscript presented in an intelligible fashion and written in standard English?

Reviewer #1: Yes

Reviewer #2: Yes

Reviewer #4: Yes

6. Review Comments to the Author

Reviewer #1: This is an interesting article. It is clear and well-written. Moreover, the suggestion that I mentioned in my previous review has already been taken into account in the revised version of the articloe. Therefore, I consider that it deserves to be published in PLOS ONE in its present form.

Reviewer #2: Dear editors,

Due to the author's revisions following my suggestions,

I recommend publishing this article.

Reviewer #4: The major comments about the revised manuscript have been clarified. My conclusion is that the revised manuscript is scientifically sound, with the main results easy to catch. The results and the subject of the paper should be of interest for researchers. I recommend the publication in Plos One.

7. PLOS authors have the option to publish the peer review history of their article (what does this mean?). If published, this will include your full peer review and any attached files.

Reviewer #1: No

Reviewer #2: No

Reviewer #4: **Yes: **Pengfei Li

---

## [Editor Report · Acceptance letter]

12 Jul 2024

PONE-D-24-22626R1 

PLOS ONE

Dear Dr. Ullah, 

I'm pleased to inform you that your manuscript has been deemed suitable for publication in PLOS ONE. Congratulations! Your manuscript is now being handed over to our production team.

Kind regards, 

on behalf of

Prof. Boris Malomed 

Academic Editor

PLOS ONE